environmental chemistry/biochemistry/inorganic chemistry

carbonic anhydrase, $CO_2$ hydration, biomimetic model complex, potentiometric pH titration, stopped-flow spectrophotometry

**Author for correspondence:**
Man Sig Lee
e-mail: lms5440@kitech.re.kr

This article has been edited by the Royal Society of Chemistry, including the commissioning, peer review process and editorial aspects up to the point of acceptance.

# Kinetic study of catalytic $CO_2$ hydration by metal-substituted biomimetic carbonic anhydrase model complexes

## DongKook Park and Man Sig Lee

Green Materials and Processes Group, Korea Institute of Industrial Technology (KITECH), 55 Jongga-ro, Jung-gu, Ulsan, Republic of Korea

DKP, 0000-0002-3058-4073

The rapid rise of the $CO_2$ level in the atmosphere has spurred the development of $CO_2$ capture methods such as the use of biomimetic complexes that mimic carbonic anhydrase. In this study, model complexes with tris(2-pyridylmethyl)amine (TPA) were synthesized using various transition metals ($Zn^{2+}$, $Cu^{2+}$ and $Ni^{2+}$) to control the intrinsic proton-donating ability. The $pK_a$ of the water coordinated to the metal, which indicates its proton-donating ability, was determined by potentiometric pH titration and found to increase in the order $[(TPA)Cu(OH_2)]^{2+} < [(TPA)Ni(OH_2)]^{2+} < [(TPA)Zn(OH_2)]^{2+}$. The effect of $pK_a$ on the $CO_2$ hydration rate was investigated by stopped-flow spectrophotometry. Because the water ligand in $[(TPA)Zn(OH_2)]^{2+}$ had the highest $pK_a$, it would be more difficult to deprotonate it than those coordinated to $Cu^{2+}$ and $Ni^{2+}$. It was, therefore, expected that the complex would have the slowest rate for the reaction of the deprotonated water with $CO_2$ to form bicarbonate. However, it was confirmed that $[(TPA)Zn(OH_2)]^{2+}$ had the fastest $CO_2$ hydration rate because the substitution of bicarbonate with water (bicarbonate release) occurred easily.

## 1. Introduction

The increased use of fossil fuels in this industrial era is causing a rapid increase in $CO_2$ concentration in the atmosphere, which contributes to global warming [1–4]. To control $CO_2$ concentration, several methods, such as $CO_2$ capture by an amine-based absorbent or carbonic anhydrase (CA) and copolymerization with an epoxide, have been studied [5–10]. Since more than one million tons per year of $CO_2$ has to be

**Scheme 1.** Proposed mechanism for the catalytic cycle of CA.

sequestered, the importance of the development of $CO_2$ capture methods has been increasing. Recently, methods using CA have been of considerable interest because the enzyme hydrates and rapidly converts $CO_2$ to carbonic acid at a rate of at least $10^6 \, M^{-1} \, s^{-1}$ [7,11,12]. However, because of the significant effects of temperature and operating pH on the catalytic activity as well as drawbacks of enzyme stability and cost, utilization of CA in an actual process is difficult [13,14]. To overcome these limitations, catalysts that mimic CA but have higher stability are currently being developed [15–18].

Most CAs have $Zn^{2+}$ at the centre of the active site, and the metal is coordinated to one water molecule and three histidine residues. The $pK_a$ of the water ligand is known to decrease from 15.7 to 7 owing to modulation by the Lewis acid, $Zn^{2+}$; thus, the hydroxide ion can be easily formed even at biological pH [19,20]. Once the nucleophilic $OH^-$ is generated, the catalytic cycle for the formation of $HCO_3^-$ in CA can be divided into three steps (scheme 1). The first step is $CO_2$ addition, in which the $Zn^{2+}$-bound hydroxide reacts with $CO_2$ to form the coordinated bicarbonate. In the second step, the bicarbonate is released by substitution with solvent water. The last step is the regeneration of the active form of CA, that is, the $Zn^{2+}$-bound water is deprotonated and the coordinated hydroxide is reformed ($pK_a = 7$) [21].

Deprotonation was reported to be the rate-determining step of the CA catalytic cycle [16,22,23]. This supports the hypothesis that the lower the $pK_a$ of the water ligand, the faster the conversion rate since deprotonation would occur more easily and bicarbonate would be more rapidly produced from $CO_2$ [24,25]. In previous studies, the development of model catalysts has been aligned mainly with the modulation of the intrinsic proton-donating ability of the water ligand in Zn catalysts to accelerate $CO_2$ hydration [26]. The synthesis of model catalysts has been promoted in this direction because it was believed that the proton-donating ability can be tuned by adding an electron-withdrawing group to the ligand or stabilizing $OH^-$ through hydrogen bonding [20,26–28]. Interestingly, the use of metal ions such as $Ni^{2+}$ and $Cu^{2+}$ was believed to adjust the internal $pK_a$ of the synthesized model complex because $pK_a$ upon hydration is lower when compared with the case of $Zn^{2+}$. Therefore, it would be possible to develop catalysts with a high hydration rate by using transition metals that are unlike those found in nature. In this paper, we synthesized CA model complexes containing $Ni^{2+}$, $Cu^{2+}$ or $Zn^{2+}$ coordinated to tris(2-pyridylmethyl)amine (TPA) and measured the $CO_2$ hydration rates by stopped-flow spectrophotometry. In addition, we determined $pK_a$, which represents the intrinsic proton-donating ability of the complex. Finally, we presented a detailed discussion of the reaction mechanism based on the experimental results.

# 2. Experimental section

## 2.1. General consideration

All reagents and solvents were obtained from commercial sources and used without further purification. All aqueous solutions were prepared using either deionized or distilled water. $^1$H-NMR (500 MHz) spectra were recorded on the Varian S500 spectrometer. Elemental analyses were performed using the Thermo Scientific FlashEA 1112 elemental analyser. Mass spectra were obtained using the Agilent 6130 mass spectrometer. Fourier-transform infrared (FT-IR) spectra were recorded on the Thermo Scientific Nicolet iS50 FT-IR spectrometer. Potentiometric measurements were carried out in the $pK_a$ mode using the Metrohm 808 Titrando titrator with the Tiamo 2.3 software and Pt1000 pH electrode. Kinetic studies were carried out in the monochromator mode using the Applied Photophysics SX20 stopped-flow spectrometer equipped with a thermoelectric temperature controller (±0.5°C).

## 2.2. Materials synthesis

[(TPA)Zn(OH$_2$)](ClO$_4$)$_2$ (**1**): an acetone solution (20 ml) of Zn(ClO$_4$)$_2 \cdot$6H$_2$O (2.0 mmol, 0.74 g) was added to an acetone solution (10 ml) of TPA (2.0 mmol, 0.58 g) under nitrogen. A white precipitate was obtained upon evaporation of the solution and subsequent washing with diethylether. Yield: *ca* 60%. $^1$H-NMR (CD$_3$OD, 500 MHz) $\delta$ = 8.68 (3H, d, pyH), 8.05 (3H, t, pyH), 7.63 (3H, d, pyH), 7.60 (3H, t, pyH). MS (ESI): *m/z* 372 ((TPA)Zn-OH$_2$), 354 ((TPA)Zn), 471 ((TPA)Zn-OH$_2$ + ClO$_4$), anal. calcd for C$_{18}$H$_{20}$Cl$_2$N$_4$O$_9$Zn: C, 37.75; H, 3.52; N, 9.78; found: C, 37.66; H, 3.42; N, 9.79.

[(TPA)Cu(OH$_2$)](ClO$_4$)$_2$ (**2**): an acetone solution (20 ml) of Cu(ClO$_4$)$_2 \cdot$6H$_2$O (2.0 mmol, 0.74 g) was added to an acetone solution (10 ml) of TPA (2.0 mmol, 0.58 g) under nitrogen. A blue precipitate was obtained upon evaporation of the solution and subsequent washing with diethylether. Yield: *ca* 60%. MS (ESI): *m/z* 371 ((TPA)Cu-OH$_2$), 353 ((TPA)Cu), 470 ((TPA)Cu-OH$_2$ + ClO$_4$), anal. calcd for C$_{18}$H$_{20}$Cl$_2$N$_4$O$_9$Cu: C, 37.87; H, 3.53; N, 9.82; found: C, 38.49; H, 3.69; N, 9.87.

[(TPA)Ni(OH$_2$)](ClO$_4$)$_2$ (**3**): an acetone solution (20 ml) of Ni(ClO$_4$)$_2 \cdot$6H$_2$O (2.0 mmol, 0.73 g) was added to an acetone solution (10 ml) of TPA (2.0 mmol, 0.58 g) under nitrogen. A dark-blue precipitate was obtained upon evaporation of the solution and subsequent washing with diethylether. Yield: *ca* 60%. MS (ESI): *m/z* 366 ((TPA)Ni-OH$_2$), 465 ((TPA)Ni-OH$_2$ + ClO$_4$), 348 ((TPA)Ni).

[(TPA)Ni(OH$_2$)](SO$_4$) (**4**): Ni(SO$_4$)$\cdot$6H$_2$O (1.0 mmol, 0.26 g) and TPA (1.0 mmol, 0.29 g) were dissolved in 10 ml of MeOH. After evaporation of the solution, the compound was crystallized in a 2 : 1 (v/v) mixture of MeOH and acetone to obtain dark-blue crystals (yield: 50%).

## 2.3. Single-crystal X-ray diffraction

A single crystal of **4** was mounted at room temperature on the tips of quartz fibres coated with Paratone-N oil and cooled under a stream of cold nitrogen. Intensity data were collected on the Bruker CCD area diffractometer, which runs the SMART software, with Mo K$\alpha$ radiation ($\lambda$ = 0.71073 Å). The structure was solved by direct methods and refined on $F^2$ using the SHELXTL software. Multi-scan absorption correction was applied using SADABS, which is part of the SHELXTL software. The structure was checked for higher symmetry by the PLATON program. Data collection and experimental details are summarized in the electronic supplementary material, table S1.

## 2.4. Potentiometric pH titration

The electrode system was calibrated with Metrohm standard buffer solutions (pH 4.00, 7.00 and 9.00) before titration. The aqueous solution of each complex (1.0 mM) was added to 2.0 mM HNO$_3$ ($I$ = 0.1 M, NaNO$_3$), and the mixture was stirred at 25°C. Titrations were performed with a standardized 0.1 M NaOH solution, and the pH was monitored to identify the half-equivalence point using the Tiamo 2.3 software.

## 2.5. Kinetic measurements using a stopped-flow spectrophotometer

The CO$_2$ hydration rate was measured using methods similar to those used previously, using the stopped-flow spectrometer [16,27]. Prior to the experiments, a saturated CO$_2$ solution was prepared by spraying deionized water with 100% CO$_2$ for at least 1 h at 25°C. Using Henry's constant, the CO$_2$ concentration of this solution was calculated to be 33.8 mM. In addition, a solution containing 0.1 M *N*-(1,1-dimethyl-2-hydroxyethyl)-3-amino-2-hydroxypropanesulfonic acid (AMPSO) buffer, 0.2 M NaClO$_4$ and 5 × 10$^{-5}$ M Tymol blue indicator (pH 9.0, adjusted with NaOH) was purged with N$_2$ gas for 1 h to remove dissolved CO$_2$. The saturated CO$_2$ solution and buffer solution were rapidly mixed at a 1 : 1 volume ratio in a stopped-flow spectrophotometer, and the absorbance at $\lambda$ = 596 nm was recorded over time to determine the uncatalysed rate. Solutions of each catalyst (1 mM TPA-M) in AMPSO buffer solution were prepared under N$_2$ atmosphere.

The initial rate $v_{int}$ of TPA-M was determined by rapidly mixing the saturated CO$_2$ solution and TPA-M solution and fitting the first 10% (10 s) of the time-dependent absorbance data to a single exponential decay function. The absorbance change was measured four times. $v_{int}$ was estimated using the equation

$$v_{int} = Q(A_0 - A_e)\left[\frac{d(\ln(A - A_e))}{dt}\right]_{t \to 0},$$

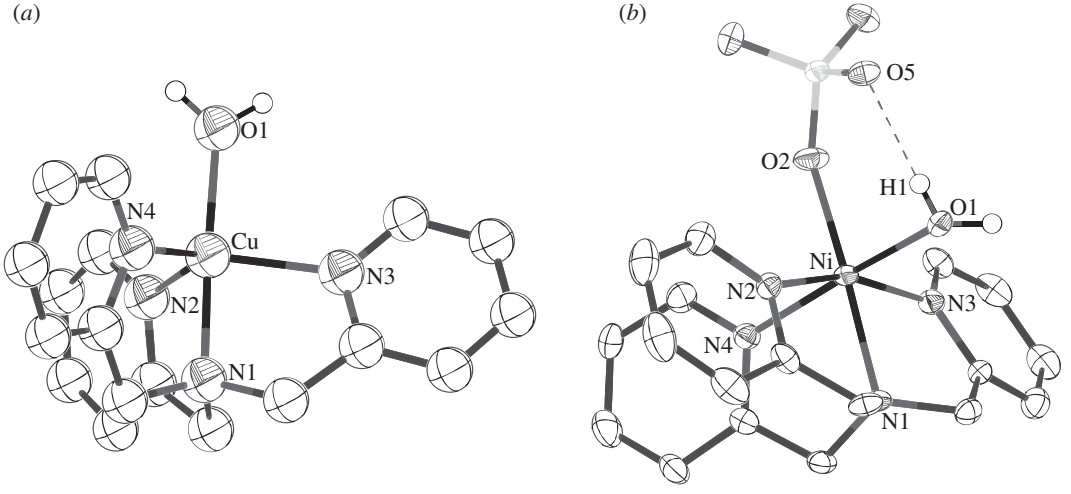

*(a)* *(b)*

**Figure 1.** Crystal structures of **2** (*a*) and **4** (*b*).

where $A_0$ and $A_e$ are the initial and final absorbance values. $Q$ is the buffer factor estimated from the AMPSO buffer solution and HCl concentration using a previously described method [16]. The rate constant $k_{obs}$ for TPA-M is the slope of the plot of $v_{init}/[CO_2]$ versus [TPA-M].

# 3. Results and discussion

To efficiently design the structure of a catalyst for the formation of $HCO_3^-$ from $CO_2$, it is very important to understand the reaction mechanism and reaction factor of the catalytic activity. One of the most important factors in controlling the reaction rate of a CA model catalyst is the structure of the surrounding ligands, which determines the electronic state of the central metal ion [18–20,29]. Thus far, several researchers have synthesized CA model compounds that contain macrocyclic or tripodal-like N4 and N3 ligands to activate $CO_2$ hydration [16,20,27,30]. Although the natural CA has an N3 ligand coordination environment around the metal ion, model catalysts that have an N4 ligand are reported to show higher catalytic activity than those that have an N3 ligand [16,31]. Recently, a computational study of the DFT wave function has shown that the N4 ligand has monodentate ground state $HCO_3^-$ binding modes with Zn, while the N3 ligand allows bidentate ground state $HCO_3^-$ binding modes [16]. This shows that the N4 ligand has better catalytic efficiency than that of the N3 ligand because of the lower dissociation energy of $HCO_3^-$ ($HCO_3^-$ release step) [32]. Moreover, compared with the N3 ligand, the N4 ligand is less likely to dimerize during the formation of the metal complex. This can be beneficial in investigating the reaction mechanism and factors [20]. Finally, an N4 ligand containing pyridine shows higher water solubility compared to that of another N4 ligand containing benzimidazole, which has been widely used in the synthesis of CA model catalysts [16,27]. Therefore, we synthesized CA model catalysts with various metal ions using the pyridine-containing N4 ligand, TPA and investigated the influence of the metal ion on the activation of $CO_2$ hydration.

## 3.1. Characterization of model complexes

The structure of a model catalyst with the $[(TPA)M-OH_2]^{2+}$ motif is critical to understand the acidity of the catalyst. Currently, structures of **2** containing a copper ion have been reported [33], but not those of **1** and **3**. Although the crystal structure of **1** has been obtained, specific information is missing [34]. Various $Zn^{2+}$ complexes containing $Zn(ClO_4)_2$, $Zn(SO_4)$ and $Zn(NO_3)_2$ were crystallized to obtain information on the structure of compound **1**; however, crystallization unfortunately yielded only thin, needle-shaped crystals. To confirm the structure of compound **3**, X-ray diffraction of compound **4**, which has $SO_4^{2-}$ instead of $ClO_4^-$, was performed. The crystal structures of **2** and **4** are shown in figure 1, and the selected bond distances and bond angles are listed in table 1 [35]. The structure of **4** shows Ni hexacoordinated with TPA, water and $SO_4^{2-}$. The bond length of three $Ni\cdots N_{pyridine}$ bonds is narrowly distributed from 2.056 to 2.082 Å, while the bond length of the fourth $Ni\cdots N_{pyridine}$ bond is slightly longer at 2.102(3) Å. In the crystal structure of $[(TBA)Zn-OH_2]^{2+}$ [TBA = tris(2-benzimidazolylmethyl)amine], which contains benzimidazole instead of pyridine, the bond length of

**Table 1.** Selected bond lengths (Å) and bond angles (°).

| [(TPA)Cu(OH$_2$)](ClO$_4$)$_2$ (**2**) | | | |
|---|---|---|---|
| Cu–O1 | 1.980(5) | Cu–N1 | 1.941(6) |
| Cu–N2 | 2.054(6) | Cu–N3 | 2.038(7) |
| Cu–N4 | 2.069(6) | | |
| O1–Cu–N1 | 177.8(2) | O1–Cu–N2 | 94.5(2) |
| O1–Cu–N3 | 98.9(2) | O1–Cu–N4 | 99.4(2) |
| N1–Cu–N2 | 83.4(2) | N1–Cu–N3 | 81.7(3) |
| N1–Cu–N4 | 82.1(2) | N2–Cu–N3 | 119.9(2) |
| N2–Cu–N4 | 118.2(3) | N3–Cu–N4 | 116.8(2) |
| [(TPA)Ni(OH$_2$)](SO$_4$) (**4**) | | | |
| Ni–O1 | 2.094(2) | Ni–O2 | 2.045(3) |
| H1–O5 | 1.97(7) | Ni–N1 | 2.102(3) |
| Ni–N2 | 2.082(3) | Ni–N3 | 2.056(3) |
| Ni–N4 | 2.076(3) | | |
| O1–Ni–O2 | 92.74(13) | O1–Ni–N1 | 92.73(13) |
| O1–Ni–N2 | 87.84(11) | O1–Ni–N3 | 92.87(11) |
| O1–Ni–N4 | 171.72(12) | O1–Ni–N2 | 87.81(11) |
| O2–Ni–N1 | 173.57(11) | O2–Ni–N2 | 101.58(13) |
| O2–Ni–N3 | 95.84(14) | O2–Ni–N4 | 92.57(12) |
| N1–Ni–N2 | 81.99(13) | N1–Ni–N3 | 80.53 |
| N1–Ni–N4 | 82.37(12) | N2–Ni–N3 | 162.52(12) |
| N2–Ni–N4 | 843.87(12) | N3–Ni–N4 | 92.90(12) |

**Table 2.** Observed pK$_a$ of complexes **1**, **2** and **3**.

| complexes | 1 | 2 | 3 |
|---|---|---|---|
| pK$_a$ | 8.0 | 7.6 | 6.0 |

Zn···N$_{amine}$ bond is definitely long compared with that of the Zn···N$_{bezimidazole}$ bond; thus, it has been claimed that TBA acts as an N3 ligand despite being an N4 ligand [27,30]. Similarly, an N4 ligand with a central tertiary amine bound to a metal ion may serve as a pseudo-N3 ligand when it is observed to have a long M···N$_{amine}$ bond length [36]. The distance between the Ni ion and water molecule is the typical Ni···O bond length of 2.094(2) Å, and the water molecule has a strong intramolecular hydrogen bonding interaction with SO$_4^{2-}$.

The presence of a water molecule in complexes **1**, **2** and **3** was also confirmed by studying the O–H vibration (electronic supplementary material, figure S1). At around 1610 cm$^{-1}$, all complexes have a sharp band that is consistent with the bending vibration of the OH group. The stretching vibration of water, typically observed at around 3300 cm$^{-1}$, appears at 3414 and 3232 cm$^{-1}$ for **2** and 3232 cm$^{-1}$ for complex **3**. Unusually, **1**, which is a Zn complex, has two vibration peaks; one at 3335 cm$^{-1}$ and other at 3256 cm$^{-1}$. Compared with the O–H vibration of a free water molecule (3506 cm$^{-1}$), a water molecule coordinated to a metal, such as in complexes **1**, **2** and **3**, is commonly observed at low frequency [37].

In CO$_2$ hydration by CA, deprotonation is a very important step that determines the reaction rate. Therefore, the acidity, which is relevant to the deprotonation step of CA model catalysts **1**, **2** and **3**, is a fundamental parameter that indicates the ease of formation of nucleophilic hydroxide ion at neutral pH. The acidity of [(TPA)M-OH$_2$]$^{2+}$ was measured by potentiometric pH titration, and the results are shown in table 2.

**Table 3.** Experimentally observed rate constant $k_{obs}$, pH-independent rate constant $k_{ind}$ and standard deviation (units are $M^{-1}\,s^{-1}$).

| complex | $k_{obs}$ | $k_{ind}$ | $\sigma_{obs}$ | $\Delta k$ |
|---|---|---|---|---|
| **1** | 645.7 | 710.3 | 14.6 | 64.6 |
| **2** | 526.4 | 527.0 | 34.8 | 0.6 |
| **3** | 542.3 | 563.9 | 4.3 | 21.6 |

The pK$_a$ of **1** is 8.0, as previously reported [38], while those of **2** and **3** are 7.6 and 6.0, respectively. These results are consistent with previous studies showing that water ionization increases in the order Zn < Co ≪ Cu, Ni when the metal ion in CA is substituted with different first-row transition metal ions [39–41]. This trend is probably due to the difference in the acidities of the metal ions. In particular, the pK$_a$ of **3**, which contains Ni$^{2+}$, is significantly lower than those of **1** and **2**.

## 3.2. Kinetics of $CO_2$ hydration by model complexes

In $CO_2$ hydration by CA, deprotonation is known as the rate-determining step; thus, it is believed that for artificial catalysts that mimic the enzyme, a lower pK$_a$ can increase the hydration rate. To understand the effect on catalytic activity when pK$_a$ is systemically modified by changing the metal, the $k_{obs}$ values of **1**, **2** and **3** were measured. Because $CO_2$ hydration is expected to be very fast, a fast kinetic measurement method, stopped-flow spectrophotometry, was used. The results are shown in table 3 and electronic supplementary material, table S2. The chemical species that is directly active in $CO_2$ hydration is not LM-OH$_2$ but the deprotonated LM-OH$^-$. $CO_2$ hydration by LM-OH$_2$ involves the following series of reactions:

$$LM\text{-}OH_2 \xrightarrow{k_a} LM\text{-}OH^- + H^+,$$

$$LM\text{-}OH^- + CO_2 \xrightarrow{k_{obs}} LM\text{-}HCO_3^-$$

and

$$LM\text{-}HCO_3^- + H_2O \longrightarrow LM\text{-}OH_2 + HCO_3^-.$$

The composition ratio of LM-OH$_2$ and LM-OH$^-$ is determined by acid-base equilibrium; thus, $k_{obs}$ can be represented as $k_{ind}$ (pH-independent form) (table 3 and electronic supplementary material, figure S2).

The absorbance change $(A_0 - A_e)$ that occurs during hydration was about 0.41, and the initial $k_{obs}$, which represents 10% of the total reaction, was obtained by fitting $(A_0 - A_e)$ to a single exponential decay function. Unexpectedly, **1** has the fastest reaction rate at 645.7 $M^{-1}\,s^{-1}$.

Complex **1** has the largest difference between $k_{obs}$ and $k_{ind}$ ($\triangle k = 64.6\ M^{-1}\,s^{-1}$, 10.0% increase), while **2** and **3** have $\Delta k$ of 21.6 (4.0% increase) and 0.6 $M^{-1}\,s^{-1}$ (0.1% increase), respectively. A large $\Delta k$ indicates that the amount of LM-OH$^-$ is less than that of LM-OH$_2$ at pH 9. That is, LZn-OH$_2$, which has the highest pK$_a$, has fewer LZn-OH$^-$ species at pH 9 than those of LNi-OH$_2$ and LCu-OH$_2$. The subsequent reaction of LM-OH$^-$ and $CO_2$ is an acid-base reaction, and the difference in reaction rate for this step is small. When exposed to air, a dinuclear LZn-OH$^-$ forms the trinuclear complex (LZn)$_3$CO$_3$, which has a triply bridging carbonate ligand [42,43]. LCu-OH$^-$ also rapidly forms the corresponding (LCu)$_3$CO$_3$ crystals [42,44]. On the other hand, LNi-OH$^-$ forms crystals in the form of L–Ni–(μ-CO$_3$)–Ni–L on reaction with $CO_2$ [45–47]. The structure of LM-HCO$_3^-$, which is consistent with $CO_2$ addition, was reported several times with respect to $CO_2$ fixation. $CO_2$ insertion into LM-OH$^-$ occurs very fast, and the complex is also known to be capable of easily absorbing atmospheric $CO_2$ [43].

Earlier in the discussion, we addressed the pK$_a$ of various metal-TPA complexes. Many studies argue that the rate-determining step of CA is either the deprotonation of the water ligand or release of bicarbonate [16,19,22,24,25,48]. Our results from potentiometric pH titration and stopped-flow spectrophotometry suggest that bicarbonate release is the rate-determining step for LNi-OH$_2$ and LCu-OH$_2$ because these complexes have lower pK$_a$ but slower $k_{obs}$ than those of LZn-OH$_2$. LNi-OH$_2$ has the most frequent deprotonation, although bicarbonate release from the Ni$^{2+}$ site does not occur easily. On the other hand, because the pK$_a$ of LZn-OH$_2$ is relatively high, deprotonation of the water ligand does not occur as easily as it does in LNi-OH$_2$ and LCu-OH$_2$. However, the substitution of bicarbonate with solvent water can be inferred to occur considerably rapidly (scheme 2).

**Scheme 2.** Proposed catalytic cycle and rate-determining step of different model complexes.

The substitution of bicarbonate with water in CA is described by the Lindskog mechanism (oxygen transfer) [49,50]. When $Zn$-$HCO_3^-$ is formed by the reaction between $CO_2$ and $Zn$-$OH^-$, bicarbonate is coordinated to Zn as a bidentate ligand. Hence, the Zn–O bond with hydroxide is cleaved and the carbonyl oxygen is newly coordinated to Zn, which is later substituted with water. In this mechanism, the mode of bicarbonate coordination to Zn is both unidentate and bidentate. However, in the case of Ni complex, substitution with solvent water would not occur easily because bicarbonate coordinated to Ni forms bidentate. Recently, the $pK_a$ of Co-substituted CA was reported to be 6.6, which is lower than the $pK_a$ of Zn-CA (6.9) [51]. However, the activity of Co-substituted CA was about three times lower than that of Zn-CA. As shown in this study, this is probably because bicarbonate release in Co-substituted CA does not occur easily.

As discussed, it is necessary to design a ligand that considers not only the active site of the natural CA but also the secondary coordination sphere in order to achieve a rate as high as that of the natural CA through the synthesis. In the case of Zn, the substitution reaction of bicarbonate and water molecules is faster than that of Cu and Ni, which suggests that mimic CA can be synthesized using Zn. However, the deprotonation reaction of Zn-bound water is slow due to high $pK_a$. To overcome this, it is possible to increase the acidity of Zn by additionally introducing an electron-withdrawing group such as the sulfonyl group into the ligand [27]. Moreover, the secondary coordination sphere of natural CA is controlled by various hydrogen bonds such as Thr-199, Glu-117, Asn-244 and Gln-92 to control the acidity of Zn [52,53]. Therefore, the catalytic efficiency of mimic CA will become closer to that of natural CA by designing a ligand to lower the $pK_a$ of Zn-bound water.

# 4. Conclusion

We have synthesized CA model catalysts with $Zn^{2+}$, $Ni^{2+}$ or $Cu^{2+}$ using TPA, a pyridine-containing N4 ligand, and investigated the effect of the metal ion on the $CO_2$ hydration rate. We observed that $pK_a$, which shows the intrinsic proton-donating ability of each complex, increased in the order **3 < 2 < 1** because of the difference in acidities of the metal ions.

The $k_{obs}$ for $CO_2$ hydration decreased in the order **1 > 3 > 2**, that is, **2**, which had relatively low $pK_a$ had the slowest $k_{obs}$. This is because while the deprotonation of water was easy, the substitution of bicarbonate with water (bicarbonate release) was difficult. In other words, in terms of $CO_2$ conversion rate, the kinetic data for the CA model catalyst with $Zn^{2+}$ implies that the advantage of the bicarbonate release step is greater than the water deprotonation step.

As with many previous CA model catalytic studies, if the intrinsic proton-donating ability can be increased by using $Zn^{2+}$, a catalyst with excellent efficiency can be developed. However, if a CA model catalyst is developed to realize various functions existing not only in the first coordination sphere (active site) of the CA but also in the secondary coordination sphere (surrounding environment), a $CO_2$ hydration rate close to that of natural CA can be realized.

Data accessibility. Electronic supplementary material is available at the Dryad Digital Repository: https://doi.org/10.5061/dryad.gn8723p [35]. Crystallographic data of compound **4** was deposited at the Cambridge Crystallographic Data Centre (CCDC) with deposition numbers of CCDC 1883607.

Authors' contributions. D.K.P. carried out the design of the study, participated in data analysis and wrote the manuscript. M.S.L. gave valuable suggestions on the writing of the manuscript. All the authors have approved the manuscript.

Competing interests. We declare no competing interests.

Funding. This study has been conducted with the support of the Korea Institute of Industrial Technology as 'Development of eco-friendly chemical materials and processing for casting, JA-18-0001.'

Acknowledgements. We thank the Korea Institute of Energy Research (KIER) for stopped-flow spectrophotometry measurement.

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
