## [Reviewer comments · Royal Society Open Science]

Review History

RSOS-190407.R0 (Original submission)

Review form: Reviewer 1

Is the manuscript scientifically sound in its present form?

No

Are the interpretations and conclusions justified by the results?

No

Is the language acceptable?

Yes

Is it clear how to access all supporting data?

No

Do you have any ethical concerns with this paper?

No

Have you any concerns about statistical analyses in this paper?

No

Recommendation?

Major revision is needed (please make suggestions in comments)

Comments to the Author(s)

To authors

The Manuscript (RSOS-190407) entitled "Kinetic study of catalytic CO₂ hydration by metalsubstituted biomimetic carbonic anhydrase model complexes " by Lee et al. described the the synthesis of model complexes with tris(2-pyridylmethyl)amine (TPA) and using various transition metals (Zn²⁺, Cu²⁺ and Ni²⁺) to control the intrinsic proton-donating ability. The effect of pK_a on the CO₂ hydration rate was investigated by stopped-flow spectrophotometry. The manuscript is interesting piece of work and is well written. I would like to recommend publication after major revision for the following reasons.

(1) The data in the manuscript were not enough to tell a interesting and complete story. Instead of putting data into different papers so to publish more papers, one really should tell a complete and interesting story with enough data, solid evidence and plausible reasoning.

(2)The way to put those data together and to analyze the data is pretty routine and boring.

(3)The authors might to answer following questions when writing the manuscript:

What is something new and exciting about the manuscript? New observations? New findings? New theory or mechanism? What would be major impact to the people working in the field?

Review form: Reviewer 2

Is the manuscript scientifically sound in its present form?

Yes

Are the interpretations and conclusions justified by the results?

Yes

Is the language acceptable?

Yes

Is it clear how to access all supporting data?

Yes

Do you have any ethical concerns with this paper?

No

Have you any concerns about statistical analyses in this paper?

No

Recommendation?

Accept as is

Comments to the Author(s)

The manuscript (ID: RSOS-190407), title "Kinetic study of catalytic CO₂ hydration by metal-substituted biomimetic carbonic anhydrase model complexes" is very interesting and suitable for

publication. The synthesized complexes were characterized and their use to mimic carbonic anhydrase was carefully discussed. The work is well described and will add knowledge to the scientific area.

Decision letter (RSOS-190407.R0)

22-May-2019

Dear Dr Park:

Title: Kinetic study of catalytic CO₂ hydration by metal-substituted biomimetic carbonic anhydrase model complexes
Manuscript ID: RSOS-190407

The editor assigned to your manuscript has now received comments from reviewers. We would like you to revise your paper in accordance with the referee and Subject Editor suggestions which can be found below (not including confidential reports to the Editor). Please note this decision does not guarantee eventual acceptance.

Please submit your revised paper before 14-Jun-2019. Please note that the revision deadline will expire at 00.00am on this date. If we do not hear from you within this time then it will be assumed that the paper has been withdrawn. In exceptional circumstances, extensions may be possible if agreed with the Editorial Office in advance. We do not allow multiple rounds of revision so we urge you to make every effort to fully address all of the comments at this stage. If deemed necessary by the Editors, your manuscript will be sent back to one or more of the original reviewers for assessment. If the original reviewers are not available we may invite new reviewers.

Please also include the following statements alongside the other end statements. As we cannot publish your manuscript without these end statements included, if you feel that a given heading is not relevant to your paper, please nevertheless include the heading and explicitly state that it is not relevant to your work.

- Acknowledgements

- Funding statement

Please include a funding section after your main text which lists the source of funding for each author.

On behalf of the Subject Editor Professor Anthony Stace and the Associate Editor Professor John Moses.

RSC Associate Editor:
Comments to the Author:
(There are no comments.)

RSC Subject Editor:
Comments to the Author:
(There are no comments.)

Reviewers' Comments to Author:
Reviewer: 1

Comments to the Author(s)
To authors

The Manuscript (RSOS-190407) entitled "Kinetic study of catalytic CO₂ hydration by metalsubstituted biomimetic carbonic anhydrase model complexes " by Lee et al. described the the synthesis of model complexes with tris(2-pyridylmethyl)amine (TPA) and using various transition metals (Zn²⁺, Cu²⁺ and Ni²⁺) to control the intrinsic proton-donating ability. The effect of pK_a on the CO₂ hydration rate was investigated by stopped-flow spectrophotometry. The manuscript is interesting piece of work and is well written. I would like to recommend publication after major revision for the following reasons.

(1) The data in the manuscript were not enough to tell a interesting and complete story. Instead of putting data into different papers so to publish more papers, one really should tell a complete and interesting story with enough data, solid evidence and plausible reasoning.

(2)The way to put those data together and to analyze the data is pretty routine and boring.

(3)The authors might to answer following questions when writing the manuscript:

What is something new and exciting about the manuscript? New observations? New findings? New theory or mechanism? What would be major impact to the people working in the field?

Reviewer: 2

Comments to the Author(s)

The manuscript (ID: RSOS-190407), title “Kinetic study of catalytic CO₂ hydration by metal-substituted biomimetic carbonic anhydrase model complexes” is very interesting and suitable for publication. The synthesized complexes were characterized and their use to mimic carbonic anhydrase was carefully discussed. The work is well described and will add knowledge to the scientific area.

Author's Response to Decision Letter for (RSOS-190407.R0)

See Appendices A & B.

RSOS-190407.R1 (Revision)

Review form: Reviewer 1

Is the manuscript scientifically sound in its present form?

Yes

Are the interpretations and conclusions justified by the results?

Yes

Is the language acceptable?

Yes

Do you have any ethical concerns with this paper?

No

Recommendation?

Accept as is

Comments to the Author(s)

Sincere the authors did their best addressing the problems brought up by referees in review process.

I would recommend it to be published at this stage.

Decision letter (RSOS-190407.R1)

01-Jul-2019

Dear Dr Park:

Title: Kinetic study of catalytic CO₂ hydration by metal-substituted biomimetic carbonic anhydrase model complexes
Manuscript ID: RSOS-190407.R1

It is a pleasure to accept your manuscript in its current form for publication in Royal Society Open Science. The chemistry content of Royal Society Open Science is published in collaboration with the Royal Society of Chemistry.

RSC Associate Editor:
Comments to the Author:
(There are no comments.)

RSC Subject Editor:
Comments to the Author:
(There are no comments.)

Reviewer(s)' Comments to Author:
Reviewer: 1

Comments to the Author(s)
Sincere the authors did their best addressing the problems brought up by referees in review process.
I would recommend it to be published at this stage.

Appendix A

Dear Editor:

I wish to submit a research paper for publication in Royal Society Open Science, titled "Kinetic study of catalytic CO₂ hydration by metal-substituted biomimetic carbonic anhydrase model complexes." The paper was coauthored by Man sig Lee. This manuscript was also proposed to be submitted to Royal Society Open Science by RSC Advances.

This study investigated the effect of the metal ion on the CO₂ hydration rates of [(TPA)M(OH₂)]₂⁺ [M = Zn, Cu, Ni; TPA = tris(2-pyridylmethyl)amine] complexes that mimic carbonic anhydrase. Water deprotonation is the rate-determining step of the zinc-containing enzyme or model complex. However, kinetic and pKa studies showed that owing to the lower pKa of the Cu and Ni complexes, bicarbonate release becomes rate determining and the reaction rate decreases. We believe that our findings significantly contribute to an understanding of the mechanism of CO₂ hydration by biomimetic complexes and factors that influence their catalytic activity.

Further, we believe that this paper would be suitable to your journal because it presents a fundamental study on an important reaction that is highly relevant to the development of approaches to mitigate the increasing CO₂ level in the atmosphere. Thus, we anticipate that this paper would be of interest not only to physical and inorganic chemists, but also environmental scientists.

This manuscript has not been published or presented elsewhere in part or in entirety and is not under consideration by another journal. We have read and understood your journal's policies, and we believe that neither the manuscript nor the study violates any of these. There are no conflicts of interest to declare.

Thank you for your consideration. I look forward to hearing from you.

Appendix B

Responses to reviewer 1

Comments to the Author(s)

To authors

The Manuscript (RSOS-190407) entitled "Kinetic study of catalytic CO₂ hydration by metal-substituted biomimetic carbonic anhydrase model complexes" by Lee et al. described the synthesis of model complexes with tris(2-pyridylmethyl)amine (TPA) and using various transition metals (Zn²⁺, Cu²⁺ and Ni²⁺) to control the intrinsic proton-donating ability. The effect of pK_a on the CO₂ hydration rate was investigated by stopped-flow spectrophotometry.

The manuscript is an interesting piece of work and is well written. I would like to recommend publication after major revision for the following reasons.

(1) The data in the manuscript were not enough to tell an interesting and complete story. Instead of putting data into different papers so to publish more papers, one really should tell a complete and interesting story with enough data, solid evidence and plausible reasoning.

- We have added a variety of papers to support our work and have made ample reasoning possible.

(2) The way to put those data together and to analyze the data is pretty routine and boring.

- Since the carbon dioxide hydration of the carbonic anhydrase model complexes is very rapid, we measured the rate using stopped-flow spectrophotometry and tried to explain the results interestingly with pK_a.

(3) The authors might answer following questions when writing the manuscript:

What is something new and exciting about the manuscript? New observations? New findings? New theory or mechanism? What would be the major impact to the people working in the field?

- We observed that the model complexes synthesized with Zn show higher pK_a than the Cu

and Ni complexes. Therefore, Zn model compound is expected to have slower carbon dioxide hydration rates than other metals, but the fastest hydration reaction was measured. This revealed that the Zn model compound is a slow reaction in the deprotonation step and this step is the rate determining step.

Our study suggests that when synthesizing and designing CA model compounds, Zn ions should be used instead of other metals and synthesized considering not only the active sites coordinated to Zn but also the secondary coordination structure.